# A MIXTURE OF VARIATIONAL AUTOENCODERS FOR DEEP CLUSTERING

## ABSTRACT

In this study, we propose a deep clustering algorithm that utilizes a variational autoencoder (VAE) framework with a multi encoder-decoder neural architecture. This setup enforces a complementary structure that guides the learned latent representations towards a more meaningful space arrangement. It differs from previous VAE-based clustering algorithms by employing a new generative model that uses multiple encoder-decoders. We show that this modeling results in both better clustering capabilities and improved data generation. The proposed method is evaluated on standard datasets and is shown to outperform state-of-the-art deep clustering methods significantly.

## 1 INTRODUCTION

Clustering is one of the most fundamental techniques used in unsupervised machine learning. It is the process of classifying data into several classes without using any label information. In the past decades, a plethora of clustering methods have been developed and successfully employed in various fields, including computer vision (Jolion et al., 1991), natural language processing (Ngomo & Schumacher, 2009), social networks (Handcock et al., 2007) and medical informatics (Gotz et al., 2011). The most well-known clustering approaches include the traditional $k$-means algorithm and the *generative model*, which assumes that the data points are generated from a Mixture-of-Gaussians (MoG), and the model parameters are learned via the Expectation-Maximization (EM) algorithm. However, using these methods over datasets that include high-dimensional data is problematic since, in these vector spaces, the inter-point distances become less informative. As a result, the respective methods have provided new opportunities for clustering (Min et al., 2018). These methods incorporate the ability to learn a (non-linear) mapping of the raw features in a low-dimensional vector space that hopefully allow a more feasible application of clustering methods. Deep learning methods are expected to automatically discover the most suitable non-linear representations for a specified task. However, a straightforward implementation of "deep" $k$-means algorithm by jointly learning the embedding space and applying clustering to the embedded data, leads to a trivial solution, where the data feature vectors are collapsed into a single point in the embedded space, and thus, the $k$ centroids are collapsed into a single spurious entity. For this reason, the objective function of many deep clustering methods is composed of both a clustering term computed in the embedded space and a regularization term in the form of a reconstruction error to avoid data collapsing.

One broad family of successful deep clustering algorithms, which was shown to yield state-of-the-art results, is the generative model-based methods. Most of these methods are based on the Variational Autoencoder framework (Kingma & Welling, 2014), e.g., Gaussian Mixture Variational Autoencoders (GMVAE) (Dilokthanakul et al., 2016) and Variational Deep Embedding (VaDE). Instead of using an arbitrary prior to the latent variable, these algorithms proposed using specific distributions that will allow clustering at the bottleneck, such as MoG distributions. This design results in a VAE based training objective function that is composed of a significant reconstruction term and a second parameter regularization term, as discussed above. However, this objective seems to miss the clustering target since the reconstruction term is not related to the clustering, and actual clustering is only associated with the regularization term optimization. This might result in inferior clustering performance, degenerated generative model, and stability issues during training.

We propose a solution to alleviate the issues introduced by previous deep clustering generative models. To that end, we propose the $k$-**D**eep **V**ariational **A**uto **E**ncoders (dubbed $k$-DVAE). Our

$k$-DVAE improves upon the current state-of-the-art clustering methods in several facets: (1) A novel model that outperforms the current methods in terms of clustering accuracy. (2) A novel Variational Bayesian framework to balance the data reconstruction and actual clustering that differs from the previous methods. (3) A network architecture that allows better generative modeling and thus more accurate data generation. Importantly, this architecture uses a lower amount of parameters compared to previous models. We implemented the $k$-DVAE algorithm on various standard document and image corpora and obtained improved results for all the datasets we experimented with compared to state-of-the-art clustering methods.

## 2 RELATED WORK

Deep clustering has been studied extensively in the literature. The most common deep clustering methods aim to project the data into a non-linear, low-dimensional feature space, where the task of clustering appears to be feasible. Then, traditional clustering methods are further applied to perform the actual clustering. Previous works have employed autoencoders (Yang et al., 2016; Ghasedi Dizaji et al., 2017; Yang et al., 2017; Fogel et al., 2019; Opochinsky et al., 2020), Variational Autoencoders (VAEs) (Jiang et al., 2016; Dilokthanakul et al., 2016; Yang et al., 2019; Li et al., 2019) and Generative Adversarial Networks (GANs) (Springenberg, 2015; Chen et al., 2016). IMSAT (Hu et al., 2017), is another recent method that augmented the training data. Our method does not make any use of augmented data during training and therefore, we do not consider IMSAT to be an appropriate or fair baseline for comparison. Additionally, the GMVAE method has shown to yield inferior performance results compared to the rest of VAE-based deep clustering, hence we do not present it in our evaluations.

Among the aforementioned work, VaDE (Jiang et al., 2016) and $k$-DAE (Opochinsky et al., 2020) are most relevant to our work. Both VaDE and our work utilize the Varitional Bayes framework, and use a probabilistic generative process to determine the data generation model. Yet, the difference lies in both the generative process and the use of several autoencoders: our network consists of a set of $k$ autoencoders, where each specializes on encoding and reconstructing a different cluster. The $k$-DAE architecture consists of a set of $k$ autoencoders, but does not consider generative modelling, which as we show, proved to be more powerful and yields significant clustering performance results in recent years.

The recent, state-of-the-art DGG method (Yang et al., 2019) was built on the foundations of VaDE, and integrates graph embeddings that serves as a regularization over the VaDE objective. Using the DGG revised objective, each pair of samples that are connected on the learned graph, will have similar posterior distributions, using the Jenson-Shannon (JS) divergence similarity metric. The other baselines used in this study are described in Section 4.2.

## 3 THE $k$-DVAE CLUSTERING ALGORITHM

In this section, we describe our $k$-Deep Variational Auto Encoders (dubbed $k$-DVAE). First, we formulate the generative model that our algorithm is based on. Next, we derive the optimization objective score. Then we discuss the differences between our model and previous VAE based algorithms such as VaDE (Jiang et al., 2016) and illustrate the advantages of our approach.

### 3.1 GENERATIVE MODEL

In our generative modeling, we assume that the data are drawn from a mixture of VAEs, each with a standard Gaussian latent r.v., as follows:

1. Draw a cluster $y$ by sampling from $p(y = i) = \alpha_i, \qquad i = 1, ..., k$.
2. Sample a latent r.v. $\mathbf{z}$ from the unit normal distribution, $\mathbf{z} \sim \mathcal{N}(\mathbf{0}, \mathbf{I})$.
3. Sample an observed r.v. $\mathbf{x}$:
   (a) If $\mathbf{x}$ is real-valued vector: sample a data vector using the conditional distribution, $\mathbf{x}|(\mathbf{z}, y = i) \sim \mathcal{N}(\boldsymbol{\mu}_{\boldsymbol{\theta}_i}(\mathbf{z}), \boldsymbol{\Sigma}_{\boldsymbol{\theta}_i}(\mathbf{z}))$.
   (b) If $\mathbf{x}$ is binary vector: sample a data vector using the conditional distribution, $\mathbf{x}|(\mathbf{z}, y = i) \sim \text{Ber}(\boldsymbol{\mu}_{\boldsymbol{\theta}_i}(\mathbf{z}))$.

$\boldsymbol{\theta}_i$ is the stacked vector of parameters of the $i$-th neural network (NN). It formulates a *decoder* NN that corresponds to the $i$-th cluster, $1 \leq i \leq k$, assuming that the total number of clusters is $k$. $\boldsymbol{\mu}_{\theta_i}(\mathbf{z}), \boldsymbol{\Sigma}_{\boldsymbol{\theta}_i}(\mathbf{z})$ are computed by a decoder NN with an input $\mathbf{z}$ and parameters $\boldsymbol{\theta}_i$. We denote the parameter set of all the decoders by $\boldsymbol{\theta} = \{\theta_1, ..., \theta_k\}$.

Note that the latent data representation $\mathbf{z}$ is drawn independently of the selected class $y$, and the class only affects when selecting the sample $\mathbf{x}$.

## 3.2 Learning the model parameters by optimizing a variational lower bound

Direct optimization of the likelihood function:

$$p(\mathbf{x};\boldsymbol{\theta}) = \sum_y \int_z p(\mathbf{z})p(y)p(\mathbf{x}|\mathbf{z}, y;\boldsymbol{\theta})d\mathbf{z}$$

is intractable. Instead, we can use variational approximation methods and learn the model parameters by maximizing the **E**vidence **L**ower **BO**und (ELBO) lower bound. The ELBO$(\boldsymbol{\theta}, \boldsymbol{\lambda})$ expression is given by:

$$\text{ELBO}(\boldsymbol{\theta}, \boldsymbol{\lambda}) = \sum_y \int_{\mathbf{z}} q(y, \mathbf{z}|\mathbf{x};\boldsymbol{\lambda}) \log p(\mathbf{x}|y, \mathbf{z};\boldsymbol{\theta})d\mathbf{z} - D_{\text{KL}}(q(y, \mathbf{z}|\mathbf{x};\boldsymbol{\lambda})||p(y, \mathbf{z};\boldsymbol{\theta})), \quad (1)$$

where $D_{\text{KL}}$ is the Kullback Leibler (KL) divergence between two density functions, and $q(y, \mathbf{z}|\mathbf{x};\boldsymbol{\lambda})$ is a conditional density function parametrized by $\boldsymbol{\lambda}$.

We use an approximate conditional density $q(y, \mathbf{z}|\mathbf{x})$ that mirrors the structure of the generative model. For each cluster we define an encoder that transforms the input $\mathbf{x}$ into the latent space of that cluster:

$$q(y = i, \mathbf{z}|\mathbf{x};\lambda) = q(y = i|\mathbf{x})q(\mathbf{z}|\mathbf{x}, y = i;\lambda_i),$$

such that $q(\mathbf{z}|\mathbf{x}, y = i;\lambda_i) = \mathcal{N}(\mathbf{z};\boldsymbol{\mu}_{\lambda_i}(\mathbf{x}), \boldsymbol{\Sigma}_{\lambda_i}(\mathbf{x}))$ where $\boldsymbol{\mu}_{\lambda_i}(\mathbf{x}), \boldsymbol{\Sigma}_{\lambda_i}(\mathbf{x})$ are computed by an encoder NN with input $\mathbf{x}$ and parameter-set $\lambda_i$ and we use the notation $\boldsymbol{\lambda} = \{\lambda_1, ..., \lambda_k\}$.

The first term of the ELBO expression (1) can be written as:

$$\sum_y \int_{\mathbf{z}} q(y, \mathbf{z}|\mathbf{x};\boldsymbol{\lambda}) \log p(\mathbf{x}|y, \mathbf{z};\boldsymbol{\theta})d\mathbf{z} = \sum_i q(y = i|\mathbf{x})\mathbb{E}_{q(\mathbf{z}|\mathbf{x}, y=i;\lambda_i)} \log \mathcal{N}(\mathbf{x};\boldsymbol{\mu}_{\theta_i}(\mathbf{z}), \boldsymbol{\Sigma}_{\theta_i}(\mathbf{z})).$$

$$(2)$$

We next use Monte-Carlo sampling to approximate the expectation in Eq. (2):

$$\mathbb{E}_{q(\mathbf{z}|\mathbf{x}, y=i;\lambda_i)} \log \mathcal{N}(\mathbf{x};\boldsymbol{\mu}_{\theta_i}(\mathbf{z}), \boldsymbol{\Sigma}_{\theta_i}(\mathbf{z})) \approx \log \mathcal{N}(\mathbf{x};\boldsymbol{\mu}_{\theta_i}(\mathbf{z}), \boldsymbol{\Sigma}_{\theta_i}(\mathbf{z})), \quad (3)$$

such that $\mathbf{z}|(\mathbf{x}, y = i)$ is sampled from $\mathcal{N}(\boldsymbol{\mu}_{\lambda_i}(\mathbf{x}), \boldsymbol{\Sigma}_{\lambda_i}(\mathbf{x}))$.

Applying the chain rule for KL divergence to the second term of the ELBO expression (1), we get:

$$D_{\text{KL}}(q(y, \mathbf{z}|\mathbf{x};\boldsymbol{\lambda})||p(y, \mathbf{z};\boldsymbol{\theta})) = D_{\text{KL}}(q(y|\mathbf{x};\boldsymbol{\lambda})||p(y;\boldsymbol{\theta}))$$
$$+ \sum_i q(y = i|\mathbf{x})D_{\text{KL}}(\mathcal{N}(\boldsymbol{\mu}_{\lambda_i}(\mathbf{x}), \boldsymbol{\Sigma}_{\lambda_i}(\mathbf{x}))||\mathcal{N}(\mathbf{0}, \mathbf{I})). \quad (4)$$

We next replace the soft clustering in Eq. (3) and Eq. (4), by a hard clustering:

$$\sum_{i=1}^k q(y = i|\mathbf{x})(\log \mathcal{N}(\mathbf{x};\boldsymbol{\mu}_{\theta_i}(\mathbf{z}_i), \boldsymbol{\Sigma}_{\theta_i}(\mathbf{z}_i)) - D_{\text{KL}}(\mathcal{N}(\boldsymbol{\mu}_{\lambda_i}(\mathbf{x}), \boldsymbol{\Sigma}_{\lambda_i}(\mathbf{x}))||\mathcal{N}(\mathbf{0}, \mathbf{I}))) \quad (5)$$

$$\approx \max_i(\log \mathcal{N}(\mathbf{x};\boldsymbol{\mu}_{\theta_i}(\mathbf{z}_i), \boldsymbol{\Sigma}_{\theta_i}(\mathbf{z}_i)) - D_{\text{KL}}(\mathcal{N}(\boldsymbol{\mu}_{\lambda_i}(\mathbf{x}), \boldsymbol{\Sigma}_{\lambda_i}(\mathbf{x}))||\mathcal{N}(\mathbf{0}, \mathbf{I}))).$$

Finally, by neglecting the term $D_{\text{KL}}(q(y|\mathbf{x})||p(y;\boldsymbol{\theta}))$ (4) (or equivalently setting $q(y|\mathbf{x}) = p(y;\boldsymbol{\theta})$), we obtain the following objective for optimization:

$$\text{ELBO}(\boldsymbol{\theta}, \boldsymbol{\lambda}) \approx \max_i \{\log \mathcal{N}(\mathbf{x};\boldsymbol{\mu}_{\theta_i}(\mathbf{z}_i), \boldsymbol{\Sigma}_{\theta_i}(\mathbf{z}_i)) - D_{\text{KL}}(\mathcal{N}(\boldsymbol{\mu}_{\lambda_i}(\mathbf{x}), \boldsymbol{\Sigma}_{\lambda_i}(\mathbf{x}))||\mathcal{N}(\mathbf{0}, \mathbf{I}))\}$$
$$\text{s.t.} \quad \mathbf{z}_i \sim \mathcal{N}(\boldsymbol{\mu}_{\lambda_i}(\mathbf{x}), \boldsymbol{\Sigma}_{\lambda_i}(\mathbf{x})). \quad (6)$$

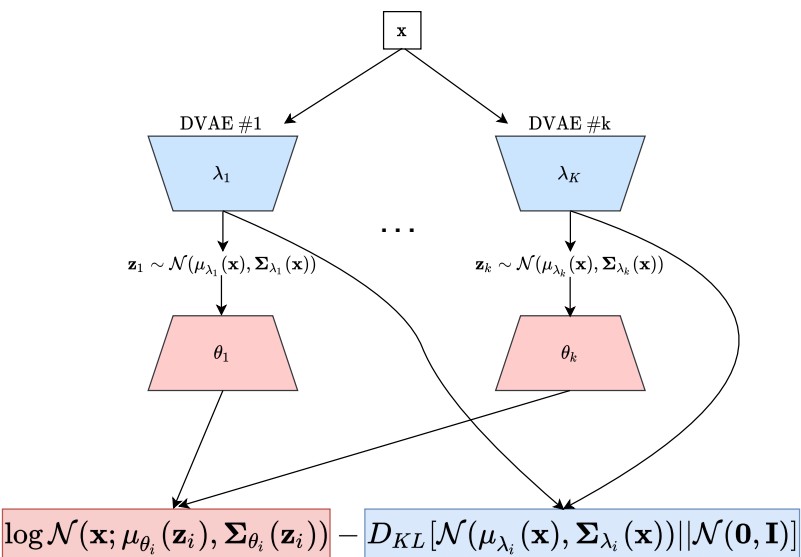

Figure 1: A block diagram of the autoencoder that computes the ELBO of the $k$-DVAE clustering method, during training phase.

---

**Algorithm 1** ELBO score computation

**Input:** Data sample $\mathbf{x}$
**Output:** Estimated score.

**for** $i = 1$ **to** $k$ **do**
    Compute $\boldsymbol{\mu}_{\lambda_i}(\mathbf{x})$ and $\boldsymbol{\Sigma}_{\lambda_i}(\mathbf{x})$ using the $i$-th encoder.
    Draw $\mathbf{z}_i \sim \mathcal{N}(\boldsymbol{\mu}_{\lambda_i}(\mathbf{x}), \boldsymbol{\Sigma}_{\lambda_i}(\mathbf{x}))$.
    Compute $\boldsymbol{\mu}_{\theta_i}(\mathbf{z}_i)$ and $\boldsymbol{\Sigma}_{\theta_i}(\mathbf{z}_i)$ using the $i$-th decoder.
**end for**
Compute the ELBO score using Eq. (6).

---

**Algorithm 2** Hard clustering

**Input:** Data sample $\mathbf{x}$
**Output:** Estimated cluster $\hat{y}(\mathbf{x})$ of $\mathbf{x}$.

**for** $i = 1$ **to** $k$ **do**
    Compute $\bar{\mathbf{z}}_i \leftarrow \boldsymbol{\mu}_{\lambda_i}(\mathbf{x})$ using the $i$-th encoder.
    Compute $\boldsymbol{\mu}_{\theta_i}(\bar{\mathbf{z}}_i)$ and $\boldsymbol{\Sigma}_{\theta_i}(\bar{\mathbf{z}}_i)$ using the $i$-th decoder.
**end for**

Compute the cluster $\hat{y}(x)$ using Eq. (8).

---

When optimizing the ELBO expression, we sample the Gaussian r.v. $\mathbf{z}_i|(\mathbf{x}, y = i)$ using the *reparameterization* trick. Note that the ELBO objective function (6) consists of a reconstruction term and a regularization term and both are involved in the clustering decision. In the derivation of the objective function above we assumed that $\mathbf{x}$ is a real-valued vector. The derivation of the ELBO objective function for the discrete case is similar. The score computation procedure is depicted in Algorithm 1 and the overall architecture of the autoencoder used in the training is depicted in Fig. 1.

### 3.3 Hard Clustering of Data Points

After the model parameters have been learned we can extract the data clustering. We chose a deterministic version of the clustering procedure (6) that avoids sampling of $\mathbf{z}$ and was empirically shown to yield more stable results in our simulations. We used the expectation vector $\bar{\mathbf{z}}_i = \boldsymbol{\mu}_{\lambda_i}(\mathbf{x})$ instead of a sampled $\mathbf{z}_i$. The hard clustering is thus defined as:

$$\hat{y}(x) = \arg\max_i (\log \mathcal{N}(x; \boldsymbol{\mu}_{\theta_i}(\bar{\mathbf{z}}_i), \boldsymbol{\Sigma}_{\theta_i}(\bar{\mathbf{z}}_i)) - \log \frac{p(\bar{\mathbf{z}}_i|x; \lambda_i)}{p(\bar{\mathbf{z}}_i; \theta_i)}) \tag{7}$$

According to our generative model $p(\bar{z}_i; \theta_i) = N(\bar{\mathbf{z}}_i; 0, I)$, and

$$\log p(\bar{\mathbf{z}}_i|\mathbf{x}; \lambda_i) = \log \mathcal{N}(\bar{\mathbf{z}}_i; \bar{\mathbf{z}}_i, \boldsymbol{\Sigma}_{\lambda_i}(\mathbf{x})) = \sum_{s=1}^{d} \log N(\bar{\mathbf{z}}_{is}; \bar{\mathbf{z}}_{is}, \Sigma_{\lambda_i}(\mathbf{x})_s) = -\sum_{s=1}^{d} \log \bar{\sigma}_{is}$$

where $d$ is the dimensionality of the latent r.v. and $\Sigma_{\lambda_i}(\mathbf{x}) = \mathrm{Var}(\mathbf{z}|\mathbf{x}, y = i) = diag(\bar{\sigma}_{i1}^2, ..., \bar{\sigma}_{id}^2)$. This finally implies:

$$\hat{y}(\mathbf{x}) = \arg\max_i(\log\mathcal{N}(\mathbf{x};\mu_{\theta_i}(\bar{\mathbf{z}}_i), \Sigma_{\theta_i}(\bar{\mathbf{z}}_i)) - \frac{1}{2}\|\bar{\mathbf{z}}_i\|^2 + \sum_{s=1}^{d}\log\bar{\sigma}_{is}) \qquad (8)$$

The hard clustering procedure is depicted in Algorithm 2.

### 3.4 COMPARISON TO THE VADE METHOD

Our method and the VaDE algorithm (Jiang et al., 2016) are both based on generative models learned by variational autoencoders. We will now briefly describe VaDE and focus on the differences from our model. The VaDE generative process is based on a MoG model combined with a non-linear function (decoder) and is given by:

1. Draw a cluster $y$ by sampling from $p(y = i) = \alpha_i, \qquad i = 1, ..., k$.
2. Sample a latent r.v. $\mathbf{z}$ using the conditional distribution, $\mathbf{z}|y = i \sim \mathcal{N}(\boldsymbol{\mu}_i(\mathbf{z}), \boldsymbol{\Sigma}_i(\mathbf{z}))$.
3. If $\mathbf{x}$ is real valued, sample it using the conditional distribution, $\mathbf{x}|\mathbf{z} \sim \mathcal{N}(\boldsymbol{\mu}_{\boldsymbol{\theta}}(\mathbf{z}), \boldsymbol{\Sigma}_{\boldsymbol{\theta}}(\mathbf{z}))$. If $\mathbf{x}$ is binary valued, sample it using the conditional distribution, $\mathbf{x}|\mathbf{z} \sim \mathrm{Ber}(\boldsymbol{\mu}_{\boldsymbol{\theta}}(\mathbf{z}))$. $\boldsymbol{\mu}_{\boldsymbol{\theta}}(\mathbf{z}), \boldsymbol{\Sigma}_{\boldsymbol{\theta}}(\mathbf{z})$ are computed by a decoder NN with an input $\mathbf{z}$ and parameters $\boldsymbol{\theta}$.

Note that unlike our method, this modeling uses the same decoder (parametrized by $\boldsymbol{\theta}$) to construct the observed data for all the different clusters. Hence the decoder is likely to be very complex. In comparing the performance of two methods in the next section, we show that much less number of parameters are needed in our model than in VaDE, and the reconstruction quality of our model is much better.

The VaDE ELBO$(\boldsymbol{\theta}, \boldsymbol{\lambda})$ term can be approximated as follows:

$$\mathrm{ELBO}(\boldsymbol{\theta}, \boldsymbol{\lambda}) \approx \log\mathcal{N}(\mathbf{x};\boldsymbol{\mu}_{\boldsymbol{\theta}}(\mathbf{z}), \boldsymbol{\Sigma}_{\boldsymbol{\theta}}(\mathbf{z}))$$
$$- \sum_{i=1}^{k} p_{\boldsymbol{\theta}}(y = i|\mathbf{z})(\underbrace{D_{KL}(\mathcal{N}(\boldsymbol{\mu}_{\boldsymbol{\lambda}}(\mathbf{x}), \boldsymbol{\Sigma}_{\boldsymbol{\lambda}}(\mathbf{x}))||\mathcal{N}(\boldsymbol{\mu}_i, \boldsymbol{\Sigma}_i))}_{C} + \log\frac{p_{\boldsymbol{\theta}}(y = i|\mathbf{z})}{\alpha_i}), \quad (9)$$

where $z$ is sampled from $\mathcal{N}(\boldsymbol{\mu}_{\boldsymbol{\lambda}}(x), \boldsymbol{\Sigma}_{\boldsymbol{\lambda}}(x))$. After the VaDE parameters are learned, the soft clustering of $x$ is $p_{\boldsymbol{\theta}}(y|z)$ where $z$ is sampled from $\mathcal{N}(\boldsymbol{\mu}_{\boldsymbol{\lambda}}(x), \boldsymbol{\Sigma}_{\boldsymbol{\lambda}}(x))$. For the full derivation, we refer the reader to Jiang et al. (2016).

Note that the term $C$ in Eq. (9) refers to the actual MoG-based soft clustering performed by VaDE during the learning phase. The clustering is thus performed here only within the ELBO *regularization* term. In our method, both the reconstruction and regularization parts of the ELBO term are involved in the clustering decision.

Another variant of our algorithm is a non-generative approach that do not have a regularization term, and it only minimizes the reconstruction error (Opochinsky et al., 2020). We show in the next section that this results in significant degradation of the clustering performance. Hence, it is required that both the reconstruction term and the regularization term of the ELBO should be involved in the clustering process.

## 4 EXPERIMENTS AND RESULTS

In this section, we present the datasets, hyperparameters, and experiments conducted to evaluate our approach's clustering results and compare it to other clustering methods.

### 4.1 DATASETS

We used the following datasets in our experiments:

**MNIST**: The MNIST dataset consists of $70,000$ handwritten (ten) digits images, of size $28 \times 28$ pixels. Prepossessing includes centering the pixel values and flattening each image to a $784$-dimensional vector.

**STL-10**: The STL-10 dataset consists of RGB colored images of size $96 \times 96$ pixels. This dataset contains a total number of 10 classes. Since clustering directly from raw pixels of high-resolution images is rather difficult, Prepossessing includes features extraction by passing the images to a pre-trained ResNet-50 (He et al., 2016) and then applying an average pooling operation to reduce the dimensionality to $2048$.

**REUTERS**: The REUTERS dataset consists of $10,000$ English news stories that relate to a total number of 4 categories. Prepossessing includes computing of 2000-dimensional TF-IDF feature vectors for the most frequent words in the articles.

**HHAR**: The Heterogeneity Human Activity Recognition (HHAR) dataset consists of $10,200$ sample records, where each sample relates to one of 6 different categories. Each sample in this dataset is a $561$-dimensional vector.

Note that we set $k$ to be the actual number of classes of the given datasets during our simulations. The overall datasets statistics are summarized in Table 1.

## 4.2 EVALUATED MODELS

We compared our method to the following state-of-the-art deep clustering algorithms:

**Autoencoder followed by Gaussian Mixture Model (AE+GMM):** This method trains a single AE using the reconstruction objective, and then applies GMM-based clustering on the embedding space.

**Variational Deep Embedding (VaDE):** Introduces a VAE based generative model that assumes the latent variables follows a mixture of Gaussians, where the means and variances of the Gaussian components are trainable (Jiang et al., 2016).

**Latent Tree Variational Autoencoder (LTVAE):** A VAE based model that assumes a tree structure of the latent variables (Li et al., 2019).

**Deep clustering via a Gaussian mixture VAE with Graph embedding (DGG):** A recent VAE based model that assumes a tree structure of the latent variables (Yang et al., 2019).

**$k$-Deep-AutoEncoder ($k$-DAE):** This algorithm uses $k$-AEs for deep clustering, where $k$ is assumed to be the number of clusters (Opochinsky et al., 2020). This method serves as the ablation study for our method, since it induces the same (reconstruction) objective without the KL term (which stands for regularization).

**$k$-Deep-Variational AutoEncoder ($k$-DVAE):** Our clustering method.

The encoder-decoder structure used for the first four methods is the same (for a fair comparison) and is composed as follows. Each encoder network uses dense layers of sizes $D-500-500-2000-10$, and each decoder network uses dense layers of sizes $10-2000-500-500-D$. All these methods use additional mid-layers (to perform clustering). This setting and the remaining hyperparameters were taken from Jiang et al. (2016), and Yang et al. (2019). For both our method and the $k$-DAE method, the autoencoders used dense layers of sizes $D - 500 - 100 - 10$ for the encoder, and $10 - 100 - 500 - D$ for the decoder. Note that although we needed to allocate one encoder-decoder network to each cluster, the number of parameters was still drastically lower than the compared

Table 1: Datasets statistics

| Dataset | # Samples | Input Dimension | # Clusters |
|---------|-----------|-----------------|------------|
| MNIST   | $70,000$  | 784             | 10         |
| STL-10  | $13,000$  | 2048            | 10         |
| REUTERS | $10,000$  | 2000            | 4          |
| HHAR    | $10,200$  | 561             | 6          |

Table 2: Clustering accuracy results for clustering benchmarks. Best performance is bolded.

|           | MNIST          | STL-10         | Reuters        | HHAR           |
|-----------|----------------|----------------|----------------|----------------|
| AE+GMM    | $82.20 \pm 0.2$ | $79.40 \pm 0.2$ | $71.12 \pm 1.1$ | $77.74 \pm 0.1$ |
| VaDE      | $94.43 \pm 0.1$ | $85.45 \pm 0.1$ | $79.84 \pm 1.5$ | $84.49 \pm 0.1$ |
| LTVAE     | $86.32 \pm 0.1$ | $90.05 \pm 0.1$ | $80.96 \pm 1.8$ | $85.10 \pm 0.2$ |
| DGG       | $97.58 \pm 0.1$ | $90.59 \pm 0.2$ | $82.30 \pm 1.2$ | $89.04 \pm 0.1$ |
| $k$-DAE   | $96.51 \pm 0.1$ | $87.30 \pm 0.1$ | $79.92 \pm 1.1$ | $87.26 \pm 0.1$ |
| $k$-DVAE  | $\mathbf{97.87 \pm 0.1}$ | $\mathbf{91.52 \pm 0.1}$ | $\mathbf{82.74 \pm 1.0}$ | $\mathbf{90.74 \pm 0.1}$ |

methods. We tried increasing the number of parameters for each method, but it did not result in any performance gains. Each encoder network outputs mean and variance vectors that form the multivariate normal distribution. The output of the decoder is a single mean vector if the input $x$ is discrete; otherwise, it also outputs a variance vector to form the normal distribution.

In our implementation of $k$-DVAE, Similar to the DGG method (Yang et al., 2019), we first pre-train the VaDE network as initialization. Then, we set the initial clusters by applying the $k$-means clustering over the VaDE embedded space. In our case, this architecture was used only in this initialization step.

### 4.3 CLUSTERING RESULTS

Clustering performance of all the compared methods was evaluated with respect to the unsupervised clustering accuracy (ACC) measure, given by

$$\text{ACC} \triangleq \max_{m \in S_k} \frac{1}{N} \sum_{i=1}^{N} \mathbb{1}\{y_i = m(\hat{y}(\mathbf{x}_i))\}$$

where $N$ is the total number of data samples, $y_i$ is the ground-truth label that corresponds to that $\mathbf{x}_i$ sample, $\hat{y}(\mathbf{x}_i)$ is the cluster assignment obtained by the model, and $m$ ranges over the set $S_k$ of all possible one-to-one mappings between cluster assignments and labels. This measures the proportion of data points for which the obtained clusters can be correctly mapped to ground-truth classes, where the matching is based on the Hungarian algorithm (Kuhn, 1955). It lies in the range of $0$ to $1$ where one is a perfect clustering result and zero is worst.

In Table 2 we depict the quantitative clustering results over the tested benchmarks compared to clustering methods. We show the mean and average ACC clustering results over ten training sessions with different random parameter initializations. The table shows that our method outperformed the other methods in terms of accuracy. In addition, using the non-variational $k$-DAE variant yields inferior results compared to our method, which emphasizes the superiority of the variational generative framework in this setup.

### 4.4 QUALITATIVE ANALYSIS

A key modeling difference between our $k$-DVAE and recent state-of-the-art models is that our model allocates a different decoder to each cluster. We saw that this yields improved clustering results. We show below that we also gain improved generation capability. In classification tasks, it is known that discriminative methods are better than generative ones since the classes are known, and we only need to find the discriminative features. However, in clustering tasks where we need to learn the clusters, there is a tight relationship between a model's generation capabilities and its clustering performance. To gain insight into our model's data generation capabilities, we present examples of images generated by the model's generator network.

To generate an example from the $i$-th clusters, we first sample a random vector $\mathbf{z}$ from the unit normal distribution and then feed it to the $i$-th decoder network, parametrized by $\boldsymbol{\theta}_i$. The VaDE/DGG algorithm, in contrast, uses a single decoder for all the clusters. Fig. 2 illustrates the generated samples for digits 0 to 9 of MNIST by our method compared to DGG[1].Note that unlike the results shown

---

[1]VaDE has a similar generative model as DGG. Thus we choose to depict the results of DGG, which is state-of-the-art.

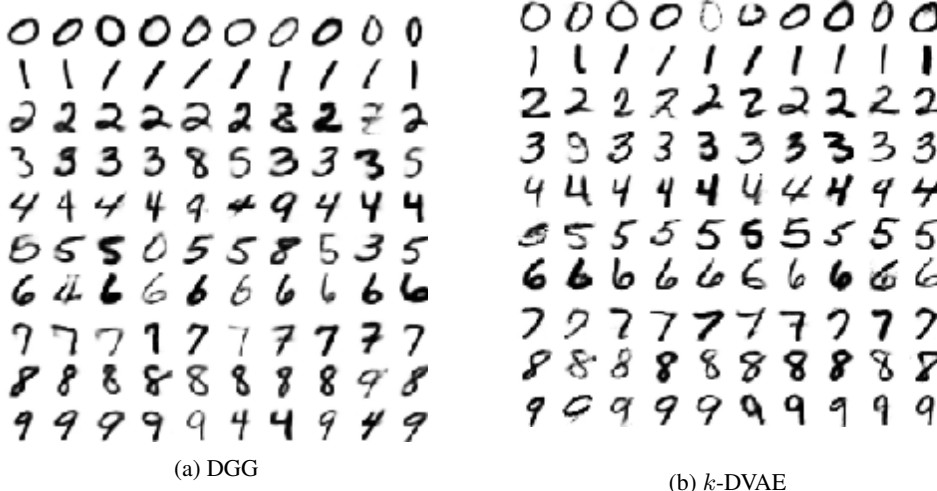

(a) DGG

(b) $k$-DVAE

Figure 2: Examples of generated digits. The digits were generated by the decoder networks of DGG and $k$-DVAE, digits in the same row come from the same cluster.

in Jiang et al. (2016), we performed the digits generation process without restricting the posterior's high values. We note in passing that in Jiang et al. (2016), the authors presented generation results only for good cases where the posterior probability of the correct clustering was at least 0.999. While both $k$-DVAE and DGG were able to generate smooth and diverse digits, the images generated by the DGG are prone to errors. In contrast, each decoder network of the $k$-DVAE successfully reconstructed its corresponding digit by only using random normal noise as an input.

## 5 CONCLUSION

In this work, we proposed $k$-Deep Variational AutoEncoder ($k$-DVAE), a neural generative model for deep clustering. This framework facilitates $k$ encoder-decoder models designed to learn insightful low-dimensional representations for better clustering. The model is optimized by maximizing the evidence lower bound (ELBO) of the data log-likelihood. Using a distinct set of $k$ parametrized models combined with the variational probabilistic framework results in a much richer representation of each cluster than previous methods. Extensive experimental results on four different datasets demonstrate our method's effectiveness over different state-of-the-art baselines, which require more parameters for training than our proposed architecture. Our qualitative analysis showcases the high quality of the generative model induced by our $k$-DVAE. Future research can extend our work by utilizing a graph embeddings similarity objective or adding a discriminator network to further regularize the posterior.

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
