# OpenReview forum: "A Mixture of Variational Autoencoders for Deep Clustering"
_ICLR.cc/2021/Conference — Reject_

### Official Review · AnonReviewer2 · 2020-10-26
**Clustering with K different VAE's, but the approximation to the ELBO is not well justified**

**Rating:** 6
**Confidence:** 4

**Review:**

The paper proposed to cluster the data using k different VAEs’. The method is different from the existing VAE-based deep clustering method (VaDE), which uses only one VAE but employs a Gaussian mixture prior to achieve the clustering goal. The difficulties of the proposed model lie at how to train the model efficiently. To this end, some approximations are made to the ELBO by using the MAP value to replace the expectation as well as dropping some KL term. The approximations are the key to the training, but not justified well. Experiments are conducted on several image and text datasets, and show superior performance comparing to existing deep clustering methods.

Strength:

1. The idea of using K different VAE’s to perform clustering is interesting, and is a good complementary to existing deep clustering methods.

2. The experimental results demonstrate the superiority of the proposed method over existing ones.

Weakness:

1. The approximation to the ELBO by dropping one KL terms is somewhat unreasonable.

2. Using MAP value to approximate the expectation is acceptable. But here, what you need is not the expectation value, but the gradient of the expectation w.r.t. model parameters \lambda. So, when you replace the MAP value with the expectation in (5), the gradient computed from the MAP expression will be much different from the exact gradient.

---

> ### Author Response · Authors · 2020-11-17
> **Response to Reviewer2**
>
> Thank you for your constructive and thoughtful comments.
> Regarding the approximations, it is not feasible to directly compute the posterior p(y=i|x). Hence we use two reasonable approximations in our derivation. A similar situation occurs in VaDE where p(y=i|x) is approximated by p(y=i|z=E[z|x]).

---

### Official Review · AnonReviewer1 · 2020-10-27
**The authors propose a variant of variational deep clustering to include K encoder-decoder neural architectures to improve the quality of the data generations and the clustering performance. The results are rather convincing and the idea is worth exploring, however the paper contains many weaknesses that should be addressed and improved.**

**Rating:** 5
**Confidence:** 4

**Review:**

1) Quality of the writing.

The paper is rather difficult to read because of many typos and confusing statements. The authors should spend more time in refining the clarity of the presentation. Some suggestions: run a spell checker, avoid sloppy statements, be more meticulous in the description of related works and improve Figure 1.
To give some examples, here are two sentences from the Introduction:

“As a result, the respective methods have provided new opportunities for clustering. These methods incorporate the ability to learn a (non-linear) mapping of the raw features in a low-dimensional vector space ...”. Here I struggle to understand which are the respective methods and which are these methods. The only models described before are the K-means and the Gaussian Mixture Model. They surely do not incorporate the ability to learn a mapping of the raw features into a latent representation.

“However, this objective seems to miss the clustering target, since the reconstruction term of is not related to the clustering and actual clustering is only associated with the regularization term optimization.”  Besides some typos, this seems a rather strong statement as VaDE and GMMVAE indeed are quite good at clustering and the fact that the clustering objective is enforced by the prior distribution in the latent space does not result in “missing the clustering target”. A reference or a more systematic description could help.


2) Hard clustering and K-DAE.
The title and the first chapters (till Chapter 3.2) are unfortunately quite misleading. The authors state that they propose a novel Variational Bayesian framework with K encoder-decoder architecture and they derive the corresponding ELBO in Chapter 3.2. However, they then choose to retain a deterministic approach. The latter is nothing else than a mixture of K Autoencoders with an L2 regularisation in the latent space. Hence, the proposed approach does not differ much from the k-DAE approach and I believe it cannot be described as a “novel Variational Bayesian framework".

3) Latent space.
One of the advantages of the VaDE method is that the latent space is nicely clustered following a Mixture of Gaussians distribution. The proposed method, on the other hand, chooses a unit normal distribution instead. As a result, the latent space cannot be used to visually investigate the data and there is no principled method to measure the uncertainty of the clustering prediction.

3) Comparison with related work.
This paper reminded me of [1] which seems quite similar. I would find the paper more persuasive if it stated what the authors do over and above.
[1] Kopf, A. et al. “Mixture-of-Experts Variational Autoencoder for clustering and generating from similarity-based representations.” ArXiv abs/1910.07763 (2019): n. pag.

---

> ### Author Response · Authors · 2020-11-17
> **Response to Reviewer1**
>
> Thank you for your constructive and thoughtful comments.
> We are afraid that the reviewer missed the main point of the paper.
> The reviewer claims that all we are doing here is a deterministic mixture of autoencoders with L2 regularization. This is NOT what we are doing. This actually appears as one of the baselines that we compared to - k-DAE. The whole point of the paper is showing that this is not the preferred way for deep clustering.
>
> We follow here the VAE paradigm. We have a generative model, and we use an amortized variational approximation for the posterior distribution implemented by a network. We further use (nondeterministic) sampling to approximate the expectation in the ELBO expression. We do not use L2 parameter regularization. The KL term in the ELBO expression (1) is commonly interpreted as a regularization.
>
> Regarding data visualization, we are interested here primarily in clustering (and data generation). If the goal would be data visualization in the latent space, we could replace the simple Gaussian model z ~  N(0,I) with a different Gaussian for each cluster, namely, z|y=i  ~  N(\mu_i,\Sigma_i).
>
> We were not aware of [1] since it is not a published paper, which seems interesting and relevant. The idea there is quite different - In contrast to our method, they use one decoder to reconstruct the samples, and the clustering is performed in the latent space, like in the VaDE architecture. Our model uses different decoders, and the reconstruction phase includes the clustering as well.

---

> > ### Comment · AnonReviewer1 · 2020-11-17
> > **Response to author**
> >
> >
> >
> > Thank you for the response and for clarifying point 2.
> > After reading the updated version of the paper, I noticed I indeed misunderstood Chapter 3.3. Hence, I increase my score. The authors state that the deterministic approach is considered after the model is trained. Maybe you could highlight that sentence to avoid potential confusion among the readers.
> > As the contributions seem a bit incremental, I would also suggest to reduce the size of Fig1 and perform experiments with more complex datasets

---

### Official Review · AnonReviewer3 · 2020-10-29
**A combination of two existing deep clustering methods with limited novelty and improvement in clustering results.**

**Rating:** 5
**Confidence:** 3

**Review:**

The paper proposes a deep clustering method based on variational autoencoders.  The proposed method adopts a k-decoders architecture with a separate decoder for each cluster.  The proposed method was shown to achieve better clustering accuracy over several other recent deep clustering methods on four real-world data sets.

The paper is good that it provides relatively detailed explanation of the proposed method.  On the other hand, the proposed method basically combines the ideas of two previous works.  It uses a variational Bayesian estimation to estimate the autoencoders in k-DAE method.  The derivation looks straightforward., and the novelty and contribution look incremental.  The paper may perhaps emphasise more on the non-trivial parts.

The proposed method has been compared with a reasonable collection of baseline methods and data sets.  Although the proposed method produced the best results on all the selected data sets, the improved clustering accuracy is relatively small (less than 1) over the best methods. Besides, the empirical results are not very interesting.  It would be better if more results could be demonstrated.  For example, could any difference between the autoencoders for the different clusters be identified in data sets other than the MNIST digit data set?

The paper does not seem to provide the source code.  So there is concern on the reproducibility of the results and whether the proposed method can be easily used by practitioners.

Overall, the proposed method has been shown have state of the art performance.  However, the paper is not very exciting as the proposed method appears to have limited novelty and the improvement in clustering accuracy is relatively small.

Minor comment:

- Figure 1 looks too large proportionally.

---

> ### Author Response · Authors · 2020-11-17
> **Response to Reviewer3**
>
> Thank you for your constructive and thoughtful comments.
>
> A VAE cost function consists of a reconstruction error term and a regularization term. Which term should be involved in clustering? VaDE (and all other VAE based clustering algorithms) used only the regularization term while k-DAE used only the reconstruction term. This paper's main novel observation is to obtain a robust clustering algorithm, and both terms should be explicitly involved in the clustering procedure.
>
> * Our source code will be made available upon publication.
> * As we stated in Section 4.2 the number of our model parameters is drastically lower than the compared methods, yet the performance results are still better. We added this clarification in the conclusion and the introduction sections.
> * In this work, we focused on a family of clustering methods that use simple neural architectures by employing only fully-connected layers, mainly for fast training and clustering capabilities. This is the motivation behind providing qualitative evaluations for the MNIST data and not for other, better-looking benchmarks. Furthermore, we employed this experiment from previous and recent papers, such as the DGG (Yang et al., 2019).

---

### Official Review · AnonReviewer4 · 2020-10-30
**Extending k-DAE with VAE**

**Rating:** 5
**Confidence:** 4

**Review:**

The authors propose to cluster a data-set into k groups using k-VAE.
The model is at the intersection of k-DAE (Opochinsky et al., 2020), and VaDE (Jiang et al., 2016).

The paper is straightforward and goes directly to the point: VAEs improve AEs.
The reason is however not discussed.

Some sentences are a bit clumsy, eg.:
"Our k(DVAE improves upon the current state-of-the-art clustering methods in several facets: (1) A novel model ...(2) A novel, "
Novelty is not per se an improvement. Besides, the third point is not novel since it is what does k-DAE.

Related Works: Why talking about augmented method?
The original paper of GMVAE suggests that it does yield SoA performances.
TYPO: 'GMVAE method has shown to yield[ed] inferior"

Figure 1 is unnecessarily large.

Section 3.4 typo: "in VaDE and the [R]econstruction quality"
To some extent, an AE is always a generative model.

Experiments: GMVAE is missing
Section 4.3 is unnecessary.

Section 4.4:
k-DVAE is also prone to error:  lines for 3, 4 and 5 contain a 9
"This obviously affects the clustering accuracy of VaDe/DGG given in Section 4.3."
DGG reports very similar ACC than k-DVAE.

"In contrast, each decoder network of the k-DVAE was able to successfully reconstruct its corresponding digit by only using y random normal noise as an input."
This is a bit far stretched: the decoder of VaDE is also able to reconstruct a number from Gaussian noise.

Typo: "by only using [y] random normal"

---

The fact that the paper is straightforward is a quality. However, some analysis are missing. For example, at first sight having k AE to  train seems a bulky situation. However the authors claim that the SoA performances can be reached with smaller architecture. How does this evolves with k? I other the words, what are the limit of VaDE and k-DVAE.
Overall, the contributions are incremental and mildly novel.

---

Indeed, the GMVAE was already omitted in DGG's paper.
Regarding section 4.3, I meant that you don't need to recall the definition of ACC.

---

> ### Author Response · Authors · 2020-11-17
> **Response to Reviewer4**
>
> Thank you for your constructive and thoughtful comments.
>
> A VAE cost function consists of a reconstruction error term and a regularization term. Which term should be involved in clustering? VaDE (and all other VAE based clustering algorithms) used only the regularization term while k-DAE used only the reconstruction term. This paper's main novel observation is that both terms should be explicitly involved in the clustering procedure to obtain a robust clustering algorithm.
>
> * We polished some sentences in the text and improved the style of writing. We also fixed the typos that were stated and others that we found.
> * Regarding GMVAE - We note in the "related work" section that GMVAE was evaluated in previous papers and yielded inferior results compared to the other baselines we evaluated. Moreover, we adopted the methods that appear in the DGG paper (Yang et al., 2019), which omitted GMVAE (but still referenced it).
> * Section 4.3 elaborates on the evaluation metric (clustering accuracy) and discusses the results from Table 1. Thus we believe we should keep it.
> * Regarding the qualitative analysis, k-DVAE is not perfect (as it makes some errors through generation) but still presents much better generation capabilities than the previous methods. We removed the sentence regarding the implication of the better generation to better accuracy.
> * Since deep clustering is considered a challenging task, standard datasets keep a low amount of ground truth clusters. The number of clusters appears in Table 1 (dataset statistics) with a varying number of clusters (4-10). Our evaluation shows that our method outperforms the baselines over all the benchmarks, independently of the number of clusters.

---

### Author Response · Authors · 2020-11-17
**General response for all the reviewers**

We thank all of you for your insightful comments. We updated the paper to account for them.

We address individual reviewers' comments in the responses to their reviews.

Note that during our rebuttal, math-mode was not available and hence the equations are addressed with regular typing.

---

### Decision · Program_Chairs · 2021-01-07
**Final Decision**

**Decision:**

Reject

**Comment:**

The submitted paper proposes a novel model/approach for deep clustering which shows good empirical performance on a set of standard benchmark datasets as compared to state of the art baseline algorithms. While I believe that this paper can be turned into a good ICLR paper, it doesn’t meet the standard of ICLR in its current form.

More specifically:
1) The quality of the writeup is poor, containing many typos but more problematically many unclear/confusing statements which are either vague/unclear, not supported by citations and/or substantiated in other parts of the paper (Examples: „This might result in inferior clustering performance, degenerated generative model, and stability issues during training.“ Or “However, this objective seems to miss the clustering target, since the reconstruction term of is not related to the clustering and actual clustering is only associated with the regularization term optimization.“). An important contribution of the paper could be to substantiate these statements and I argue that achieving better performance alone is not sufficient therefore.

2) From a theoretical perspective, it would be interesting whether there is any justification for statements like the ones references above. Furthermore, the authors’ approach involves several approximations whose implications are neither studied nor explained. The authors responded only partially to questions in that regard by reviewers leaving certain concerns unanswered.

3) From an empirical perspective, an extended study of the proposed approach would help t better understand its benefits over existing approaches. Commonly considered settings like mismatch in the number of specified clusters are not studied at all. The proposed approach also seems to be initialized by VaDe (mentioned in Section 4) and it would be interesting to understand to which extend this is necessary and why (and how does performance change/degrade if this is not done). It also makes statements regarding stability unclear as training VaDe itself can be quite challenging. Furthermore, the overall algorithm for training the proposed model should be presented in a compact form in the paper. The paper should also be self-contained in the sense of containing information on the important hyper-parameters needed for training the proposed model.

In summary, the proposed approach is potentially interesting but the paper should not be accepted in its current form.